# Editing Common Sense in Transformers

**Anshita Gupta**♠* **Debanjan Mondal**♠* **Akshay Krishna Sheshadri**♠*
**Wenlong Zhao**♠ **Xiang Lorraine Li**♣* **Sarah Wiegreffe**♡* **Niket Tandon**♡*

♠University of Massachusetts Amherst, ♣University of Pittsburgh, ♡Allen Institute for AI
{anshitagupta,debanjanmond,asheshadri,wenlongzhao}@cs.umass.edu
xianglli@pitt.edu, wiegreffesarah@gmail.com, nikett@allenai.org

## Abstract

Editing model parameters directly in Transformers makes updating open-source transformer-based models possible without re-training (Meng et al., 2023). However, these editing methods have only been evaluated on statements about encyclopedic knowledge with a single correct answer. Commonsense knowledge with multiple correct answers, e.g., an apple can be green or red but not transparent, has not been studied but is as essential for enhancing transformers' reliability and usefulness. In this paper, we investigate whether commonsense judgments are causally associated with localized, editable parameters in Transformers, and we provide an affirmative answer. We find that directly applying the MEMIT editing algorithm results in sub-par performance, and propose to improve it for the commonsense domain by varying edit tokens and improving the layer selection strategy, i.e., MEMIT$_{CSK}$. GPT-2 Large and XL models edited using MEMIT$_{CSK}$ outperform best-fine-tuned baselines by 10.97% and 10.73% F1 scores on PEP3k and 20Q datasets. In addition, we propose a novel evaluation dataset, PROBE SET, that contains unaffected and affected neighborhoods, affected paraphrases, and affected reasoning challenges. MEMIT$_{CSK}$ performs well across the metrics while fine-tuning baselines show significant trade-offs between unaffected and affected metrics. These results suggest a compelling future direction for incorporating feedback about common sense into Transformers through direct model editing.[1]

## 1 Introduction

Transformer-based language models (LMs) have achieved great success in NLP (Brown et al., 2020) but they still exhibit factual mistakes (Lewis et al., 2020; Shuster et al., 2021), commonsense mistakes

---

*Co-first and last authors. Lorraine's work done at AI2.

[1]Code and datasets for all experiments are available at https://github.com/anshitag/memit_csk

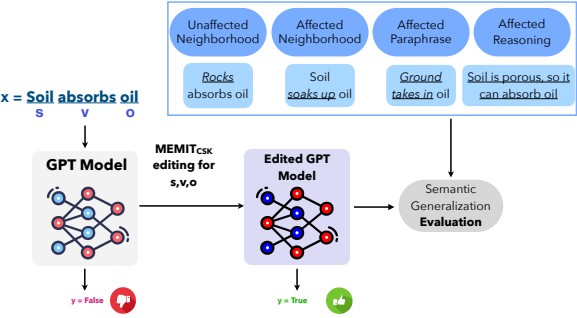

Figure 1: Proposed framework – MEMIT$_{CSK}$, for editing and evaluating plausible commonsense knowledge in Transformers. Given a plausible <Subject, Verb, Object> commonsense statement, MEMIT$_{CSK}$ edits parameters at different token and layer locations (described in §3). Edited model is evaluated for semantic generalization (depicted in dark blue box) and configuration generalization defined in §3.

(Bender and Koller, 2020; Marcus, 2021; Talmor et al., 2019; Bhargava and Ng, 2022), and consistency errors (Tam et al., 2022; Devaraj et al., 2022; Weng et al., 2020). Retraining or finetuning LMs to overcome these errors is costly and uninterpretable. To address this, prior research (Meng et al., 2022, 2023) has shown that model predictions often correlate strongly with certain neuron activations and parameter editing methods can effectively correct encyclopedic factual mistakes.

However, it remains unclear whether these editing methods can scale beyond encyclopedic facts to fix commonsense errors in Transformers. Commonsense knowledge involves more uncertainty and variation than encyclopedic knowledge. Consider a subject-verb-object triple $(s, v, o)$. In the encyclopedic domain, $s$ and $v$ often map to one "$o$", e.g., the Eiffel Tower is located in the city of "Paris". On the contrary, commonsense knowledge is harder to enumerate and $s$ and $v$ can be mapped to many "$o$", e.g., an apple has colors that can plausibly be "green", "red", "yellow", "white", and their interpolation. We aim to answer (i) *whether*

*commonsense plausibility information is **also** localized in specific hidden states of Transformers*, and if so, (ii) *can model editing on those units effectively repair incorrect commonsense plausibility judgments?*

To this end, we focus on the subject-verb-object binary plausibility classification task utilizing two commonsense datasets, 20 Questions (20Q, Porada et al. (2021)) and Physical Plausibility Commonsense (PEP3k, Wang et al. (2018)). We perform causal mediation analysis (Pearl, 2001; Vig et al., 2020; Meng et al., 2022) on GPT-2 Large and XL models and their fine-tuned checkpoints (Base-finetuned models), at various part-of-speech locations. While the zero-shot models perform poorly on the task and exhibit no causal pattern, we find clear causal associations between predictions and localized parameters at subject, verb, and object locations in the Base-finetuned models. We then investigate if we can edit relevant parameters in the Base-finetuned models to correct their mistakes. While directly applying the MEMIT editing algorithm (Meng et al., 2023) to edit subject tokens results in sub-par performances, we extend MEMIT to MEMIT$_{\text{CSK}}$ by editing various token locations and improving the edit layer selection strategy.

We demonstrate the advantage of MEMIT$_{\text{CSK}}$ compared to fine-tuning the model ("repair-finetuning")[2] from two angles: semantic generalization and configuration generalization. Semantic generalization requires that commonsense judgments are repaired while their paraphrases, neighbors, and reasoning-based queries are also answered correctly – some should be affected and others unaffected by the editing. We create a PROBE SET for 20Q and PEP3k datasets to contain efficacy, unaffected neighborhood, affected neighborhood, affected paraphrase, and affected reasoning challenges. We also evaluate configuration generalization for each method to determine whether a strategy (hyperparameter combination) picked on an EDIT VALIDATION SET can achieve good performance on a separate EDIT SET. Our proposed framework for editing and evaluating commonsense knowledge in transformers is depicted in Fig. 1.

Our contributions are five-fold. (1) We show strong causal associations between commonsense judgments and localized parameters in Base-

---

[2]We refer to fine-tuning to repair incorrect predictions as "repair-finetuning" to differentiate from the initial fine-tuning we perform to fit GPT2 for the task ("base-finetuning").

finetuned GPT-2 Large and XL models. (2) We extend the MEMIT editing algorithm to MEMIT$_{\text{CSK}}$ by varying edit tokens and improving the edit layer selection strategy, resulting in 4.58% and 1.99% F1 improvement for GPT-2 XL on EDIT VALIDATION SET of PEP3k and 20Q. (3) GPT-2 XL edited by MEMIT$_{\text{CSK}}$ outperforms repair-finetuned baselines by 10.97% and 10.73% F1 on the EDIT SET of PEP3k and 20Q, exhibiting favorable configuration generalization. (4) GPT-2 XL edited by MEMIT$_{\text{CSK}}$ performs well across the affected and unaffected metrics in our constructed PROBE SET for semantic generalization, while fine-tuned baselines exhibit significant tradeoffs between unaffected and affected metrics. (5) We show that edited models achieve clearer associations between judgments and localized parameters on previously incorrectly predicted samples, solidifying the correlation between causal analyses and performances. These results suggest a compelling future direction of incorporating feedback about common sense in transformers on the fly through direct model editing.

## 2 Background

The MEMIT (Mass Editing Memory in a Transformer) method proposed by Meng et al. (2023) demonstrates its effectiveness in editing up to 10,000 factual associations in transformer models on zsRE (Levy et al., 2017) and their proposed COUNTERFACT dataset, designed to test factual recall. We describe some background here but otherwise refer the reader to Appendix A.1 and Meng et al. (2022, 2023) for a more detailed description.

### 2.1 Causal Tracing

Given a model, the method takes a concatenation of subject $s$ and verb $v$ as input prompt $x$ and predicts the corresponding object $o$ as prediction $y$. For a correctly-predicted $(x, y)$ pair, causal tracing consists of the following three steps: **Clean run –** The input prompt is provided to the model and the predicted probability of the correct object, $\mathbb{P}[y]$, is calculated; **Corrupted run –** The subject tokens are corrupted with noise and the corresponding probability of the ground truth object, $\mathbb{P}_*[y]$, is computed; **Corrupted-with-restoration run –** The same corrupted input is given, but at a certain token $i$ and layer $l$, the model is forced to output the clean state activation from the clean run. In this setting, the probability of the correct object, $\mathbb{P}_{*, \text{clean } h_i^{(l)}}[y]$, is computed.

**Total effect (TE)** is defined as $\mathbb{P}[y] - \mathbb{P}_*[y]$, while the **indirect effect (IE)** of a specific hidden state $h_i^l$ is defined as $\mathbb{P}_{*,\,\text{clean}\,h_i^{(l)}}[y] - \mathbb{P}_*[y]$. The average total effect (**ATE**) and average indirect effect (**AIE**) are computed across multiple examples for each hidden state.

**Severed Causal Tracing:** To disentangle the impact of MLP and attention in each layer, MEMIT analyzed the effect on the attention layer by fixing the MLP output at the corrupted run value, so that it is unaffected when inserting clean state $h_i^l$. This can be viewed as severing the MLP effect when analyzing the effect on attention. Similarly, this can be done by severing attention layers.

## 2.2 Memory Editing

MEMIT identified the crucial parameters significantly impacting the model's prediction through causal tracing. They selected the layer with the highest AIE and its preceding layers as the edit layers $\mathcal{R}$. We extend MEMIT's editing strategy, described in Meng et al. (2023), to the commonsense domain.

## 3 Method

We now set out to investigate our main research question: **is commonsense plausibility information *also* localized in specific MLP hidden states of an LM**, and, if so, **can MEMIT-style editing effectively repair incorrect commonsense plausibility judgments?**

To investigate this, we conduct experiments that address important sub-questions, focusing specifically on the commonsense plausibility task (Porada et al., 2021). The task is to predict a label $y \in \{\text{True}, \text{False}\}$ given an input triple $x = (s, v, o)$. An example can be seen in Fig. 1[3].

### 3.1 Is high task performance needed to achieve a strong causal tracing result?

Because model parameter editing relies on selecting a token and layer position based on the maximum AIE, we hypothesize that model performance may impact the resulting causal tracing graph. In

---

[3]This dataset framing addresses challenges with the direct analog of the factual recall task (predicting $y = o$ given $x = (s, v)$) in the commonsense setting. Since there can be multiple correct object completions for commonsense, e.g., *rice, meat, bread* are all valid completions for the phrase *People eat _*, evaluating only the argmax completion does not rigorously assess a model's understanding of *plausibility* of events.

particular, since a model that performs near-random on a task will also perform close-to-random during a corrupted run, overall AIEs may be low as a result. This relationship has not been investigated in prior work — in contrast to the factual encyclopedic datasets used in previous studies, the zero-shot performance of language models on the commonsense plausibility task can be poor. Thus, we perform causal tracing on commonsense datasets in two experimental settings: zero-shot (Meng et al., 2022), and after fine-tuning models on plausibility tasks; we refer to this fine-tuning as **base-finetuning**.

### 3.2 Does the part of speech and model layer locations affect causal tracing conclusions and edit success?

Prior work on editing encyclopedic knowledge focuses on subject corruption and editing since factual knowledge is mostly associated with the subject and the object is directly predicted. In contrast, common sense and plausibility judgments depend on each element of the sentence. Therefore, we analyze three types of corruption and edit locations: subject, verb, and object.

MEMIT (Meng et al., 2023) edits a five-layer window whose last layer has the highest AIE in the severed causal graph. This strategy only considered the last layer effect but ignored all the other layers in the window. To mitigate this, we consider edit layers as a hyperparameter and search from a list of MEMIT's five-layer window and also the window having **max moving average of AIE**[4]. A detailed explanation of our layer selection strategy is presented in Appendix A.7. We denote our modified editing method with varying edit tokens and a more robust layer selection strategy as MEMIT$_{\text{CSK}}$.

### 3.3 Does MEMIT$_{\text{CSK}}$ exhibit configuration generalization?

Prior work on model editing tunes hyperparameters and reports performances of editing algorithms on the *same* data splits. We study *configuration generalization* – whether editing hyperparameters pre-selected on some data can be effectively transferred to an unseen data split. The motivation is that running parameter sweeps on new data points for editing can be time-consuming and costly. Since commonsense knowledge is innumerable, it is favorable if users may provide contextual feedback

---

[4]We also consider neighboring windows shifted by 1 layer, exploring windows of size 3 and 5 within this space.

to change model behaviors on the fly using pre-selected hyperparameters. We thus create an EDIT VALIDATION SET and an EDIT SET for each dataset. We select hyperparameters on the EDIT VALIDATION SET and study the transferability of the best-found setting of MEMIT$_{\text{CSK}}$ and repair-finetuning baselines to EDIT SET (§5.3).

### 3.4 Does MEMIT$_{\text{CSK}}$ exhibit semantic generalization?

It is not enough to report the success of a direct editing method on the original dataset since edit methods can (and should) have propagational effects on instances beyond the dataset (Meng et al., 2022). To compare and assess *semantic generalization* of updates, we augment incorrectly predicted samples with neighborhood instances and paraphrases that *should* be affected by an edit, similar to the prior fact editing work. We additionally include neighborhood instances that *should not* be affected. Performance on the **unaffected neighborhood** measures the update's specificity, while performance on the **affected neighborhoods** and **affected paraphrases** indicates its generalization.

Additionally, editing the plausibility of a commonsense statement should affect reasoning chains involving that statement. Entities and knowledge are interconnected, often requiring updates to one component of commonsense knowledge when modifying another. To this end, we add a fourth category of augmentations, **affected reasoning**, to test whether successful edits correct aspects of a model's commonsense reasoning. The augmentations, which form the PROBE SET, are excluded during editing and solely used for evaluation purposes. We provide examples in Fig. 1 and Table 1.[5]

### 3.5 Does MEMIT$_{\text{CSK}}$ outperform finetuning for repairing commonsense knowledge?

To answer our main research question, we compare MEMIT$_{\text{CSK}}$ applied to the MLP hidden states most strongly identified by our causal tracing experiments against **finetuning baselines**, which we refer to as **repair-finetuning**. We compare both methods' performance on edit efficacy (how many incorrect predictions are fixed), overall F1 score and relapse (how much the edit hurts by changing previously correct predictions), and semantic generalization metrics. Unlike prior work, we also investigate whether such improvements exhibit them-

selves in causal patterns by repeating the causal tracing experiments on the MEMIT$_{\text{CSK}}$-edited and repair-finetuned models, to solidify the tie between discovery and correction.

## 4 Experimental Setup

### 4.1 Models

We perform experiments on GPT-2 Large and XL (Radford et al., 2019).[6] We finetune checkpoints from Huggingface Transformers (Wolf et al., 2020) on Training Sets to obtain Base-finetuned models (Base Model), whose mistakes are then repaired by MEMIT$_{\text{CSK}}$ or repair-finetuning. The base-finetuning hyperparameters are in Appendix A.5. All predictions are made by $\arg\max_{y \in \{\text{True,False}\}} p(y|x)$ where $x$ is a commonsense subject-verb-object statement.

### 4.2 Data and Evaluation

We use Porada et al. (2021)'s versions of two commonsense plausibility datasets, PEP3k and 20Q. We build three splits from each dataset: Training Set, EDIT VALIDATION SET, and EDIT SET. Since zero-shot GPT-2 Large and XL perform poorly on PEP3k and 20Q out-of-the-box, we create Training Sets for base-finetuning the models on the task.

The Training Set and EDIT VALIDATION SET are formed by randomly dividing the validation set from Porada et al. (2021) into an 80%-20% split. The EDIT SET is created using the test set from Porada et al. (2021). Because both datasets' instances are unnatural (e.g., "man swallow paintball"), we use GPT-3 text-davinci-003 to reformat them into natural language while retaining the $(s, v, o)$ format, e.g., "A man swallows a paintball". More details and dataset statistics are in Appendix A.2.

We report three metrics on the EDIT VALIDATION SET and EDIT SET: **F1 Score** ($\uparrow$), a measure of overall performance; **Efficacy** ($\uparrow$), the percentage of previously-incorrect predictions which are corrected by an update method; and **Relapse** ($\downarrow$), the percentage of instances which were previously predicted correctly but are now predicted incorrectly following an update.

### 4.2.1 Constructing the PROBE SET

For the subset of EDIT SET that was incorrectly predicted by both GPT-2 Large and XL Base Model,

---

[5]More details about dataset construction are in §4.2.1.

[6]We report GPT-2 XL results in the main paper. GPT-2 Large results and similar findings are in the Appendix.

| Statement | Plausibility Label | Unaffected Neighborhood | Affected Neighborhood | Affected Paraphrase | Affected Reasoning |
|---|---|---|---|---|---|
| **PEP3k** | | | | | |
| Soil absorbs oil | True | **Rocks** absorbs oil
Soil absorbs **fire** | **Dirt** absorbs oil
Soil **consumes** oil
Soil absorbs **grease** | Ground takes in oil
Dirt soaks up oil
Land absorbs oil | Oil is liquid, so it spreads over surface
Soil is porous, so it can absorb oil |
| Tree kick ball | False | **House** kick ball
Tree kick **rock** | **Plant** kick ball
Tree **strike** ball
Tree kick **sphere** | Tree was used to propel a ball
Tree was used to kick a ball
Tree was used to hit a ball | Tree doesn't have legs
Legs are needed to kick ball |
| **20Q** | | | | | |
| Sunglasses block sun | True | **Trees** block sun
Sunglasses block **rain** | **Shades** block sun
Sunglasses **obscure** sun
Sunglasses block **light** | Sunglasses act as a shield from sun
Sunglasses obstruct the sun's light
Sunglasses filter out sun's brightness | Sunglasses have dark lenses
Dark lenses reduce light that enters eyes |
| Furnishings make noise | False | **Computers** make noise
Furnishings make **color** | **Fixtures** make noise
Furnishings **produce** noise
Furnishings make **sound** | Furniture can be noisy
Furniture can create sound
Furniture can be a source of noise | Furnishings are inanimate objects
Inanimate objects cannot make noise |

Table 1: Examples chosen through random sampling from the PEP3k and 20Q PROBE SET. Unaffected neighborhood samples are created by individually augmenting the **subject** and **object** with different, but relevant instances from the source statement. Likewise, affected neighborhood samples are created by individually augmenting the **subject**, **verb**, and **object** with synonymous instances from the source statement. Further details are in §4.2.1.

we augment each instance with neighborhood instances that *should* or *should not* be affected by an edit that fixes the incorrect prediction on the dataset instance using GPT-3 (details in Appendix A.9). We combine the incorrectly predicted instances from EDIT SET and the per-instance augmentations to form the PROBE SET for evaluating semantic generalization. Dataset examples are in Table 1 and statistics in Appendix A.2.

**Unaffected Neighborhood.** To evaluate the specificity of the edits, for each $\{s, v, o\}$, we generate a set of relevant but different instances $(s', v, o)$ and $(s, v, o')$ that should *not* change when $\{s, v, o\}$ is edited. The metric measures the percentage of post-update predictions $\arg\max \mathbb{P}(s', v, o)$ and $\arg\max \mathbb{P}(s, v, o')$ that remain equivalent to pre-update predictions.

**Affected Neighborhood.** To assess the impact of changes on similar meaning prompts for each $(s, v, o)$, we generate a set of synonyms as $(s', v, o)$, $(s, v', o)$ and $(s, v, o')$. The score measures the percentage of post-update predictions $\arg\max \mathbb{P}(s', v, o)$, $\arg\max \mathbb{P}(s, v', o)$ and $\arg\max \mathbb{P}(s, v, o')$ which are equal to the ground truth label for $(s, v, o)$.

**Affected Paraphrase.** To evaluate the impact on synonymous prompts, we generate a set of paraphrases as $(s', v', o')$. Since paraphrases should also be successfully edited, the metric is the percentage of post-update predictions $\arg\max \mathbb{P}(s', v', o')$ which are equal to the ground truth label for $(s, v, o)$.

**Affected Reasoning.** To assess the updated model's connectivity, we generate a two-step chain of valid reasoning prompts $\{R_1, R_2\}$. For instance,

with the phrase *"Furnishings do not make noise"*, $R_1$ could be "Furnishings are inanimate objects", and $R_2 =$ "Inanimate objects cannot make noise". The metric is the percentage of post-update predictions $\arg\max \mathbb{P}(R_1)$ and $\arg\max \mathbb{P}(R_2)$ which are equal to the *True* label.

### 4.3 Editing and Finetuning Methods

We select hyperparameters to maximize F1 on the EDIT VALIDATION SET (§3.3). For editing, we search for the edit layer range, edit token position (last $\{s, v, o\}$), and learning rate. For the repair-finetuning baseline, we search for the learning rate, batch size, and the number of epochs.

For editing, we perform causal tracing on the correctly-predicted samples of the EDIT VALIDATION SET to inform layer selection. We apply repair-finetuning and editing methods to repair incorrect predictions on EDIT VALIDATION SET, EDIT SET, and PROBE SET.

We explore two variants of repair-finetuning. **RFTFixed Epoch** uses the same exact configuration found on EDIT VALIDATION SET. We hypothesize that it is prone to overfitting due to the absence of early stopping. To maximize the potential of repair-finetuning, we analyze another variant **RFTEarly Stop**, which runs for a maximum of 10 epochs and selects the checkpoint with the highest F1 score on the entire EDIT SET. This should mitigate overfitting and reduce relapse. In contrast, the editing experiments always use the exact configuration obtained from EDIT VALIDATION SET.

## 5    Results & Discussion

### 5.1    High task performance is crucial for achieving strong causal tracing results

Zero-shot prompting produced near random accuracies (51.30% and 51.87% on the EDIT VALIDATION SET split of PEP3k and 20Q respectively for GPT-2 XL) and chaotic causal patterns with no localization as shown in Fig. 2[7]. In contrast, the Base Model exhibited significantly superior performance (77.12% on PEP3k and 73.96% on 20Q) and the resulting causal patterns were more distinct with a substantially higher AIE and strong localization. Therefore, we deduce that a significant correlation exists between high task performance and strong causal patterns, and use the Base Model for editing experiments.

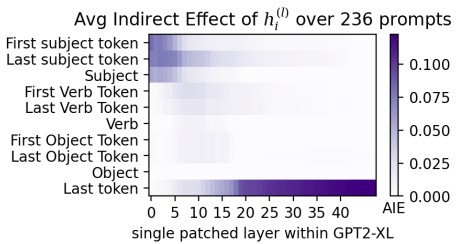

(a) Base Model with 77.12% accuracy.

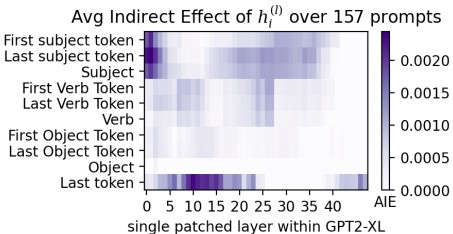

(b) Zero-shot model with 51.30% accuracy.

Figure 2: Base-finetuned vs. Zero-shot GPT-2 XL causal tracing on PEP3k EDIT VALIDATION SET. Patterns are unclear for the Zero-shot model while they are distinct for the Base Model. Consistent observations are found for the 20Q dataset (Fig. 6).

### 5.2    Targeted part of speech and layer locations affect causal tracing conclusions and edit success

As shown in Fig. 3, the last token at the later layers has a high AIE which is trivial since fixing hidden states or MLPs in those layers restores most of the required information. We also observed strong AIE

---

[7]Normalization with domain conditional Pointwise Mutual Information (PMI; Holtzman et al., 2021) did not result in any significant improvement with accuracy $PMI_{DC}$ = 52.94% on PEP3k.

at the earlier layers for the corrupted tokens. This finding is non-trivial and emphasizes the importance of earlier layers while predicting plausibility. AIE is more pronounced at the last corrupted token compared to the first corrupted token consistently across all models and datasets. Therefore, we focus on the last $(s, v, o)$ editing. Additional causal tracing results are present in Appendix A.10.

Fig. 4 compares the average AIE at last corrupted token for unmodified, severed MLP and Attention causal graphs for all edited tokens. We notice a clear gap in AIE for MLP graphs at the earlier layers. This observation aligns with previous observations in MEMIT for encyclopedic knowledge. In contrast to encyclopedic facts, we observed the highest AIE in earlier MLP layers instead of middle layers. This demonstrates the importance of earlier layers in commonsense predictions. Interestingly, in the object corruption plot, we observed a small peak at the beginning, before the highest AIE later. We thus expanded the hyperparameter space to include the initial layer windows for the object edit layers. Table 2 presents edit layers included in hyperparameter search with the max moving average of AIE, comparing windows of size 3 and 5 using different editing tokens $\{s, v, o\}$. In all cases, the max moving average resulted in a different set of layers selected than MEMIT, where the max AIE layer is used to edit 5 layers- the selected layer and the previous 4 layers.

| Model | Edit Token | Layer with Max AIE | Layers with Max Moving Average AIE | |
|---|---|---|---|---|
| | | | Window=3 | Window=5 |
| GPT-2 Large | Last Subject | 8 | 8,9,10 | 8,9,10,11,12 |
| GPT-2 Large | Last Verb | 4 | 4,5,6 | 4,5,6,7,8 |
| GPT-2 Large | Last Object | 12 | 11,12,13 | 10,11,12,13,14 |
| GPT-2 XL | Last Subject | 5 | 4,5,6 | 2,3,4,5,6 |
| GPT-2 XL | Last Verb | 5 | 5,6,7 | 3,4,5,6,7 |
| GPT-2 XL | Last Object | 12 | 10,11,12 | 9,10,11,12,13 |

Table 2: Layer with max AIE and set of layers with max moving average AIE for the PEP3k EDIT VALIDATION SET

These two changes resolve in MEMIT$_{CSK}$. Table 3 compares original MEMIT[8] (only subject edit with fixed edit layers) with the best-performing edit of MEMIT$_{CSK}$ on EDIT VALIDATION SET. MEMIT$_{CSK}$ consistently outperforms MEMIT across datasets and models.

---

[8]Detailed results are in Appendix A.4.

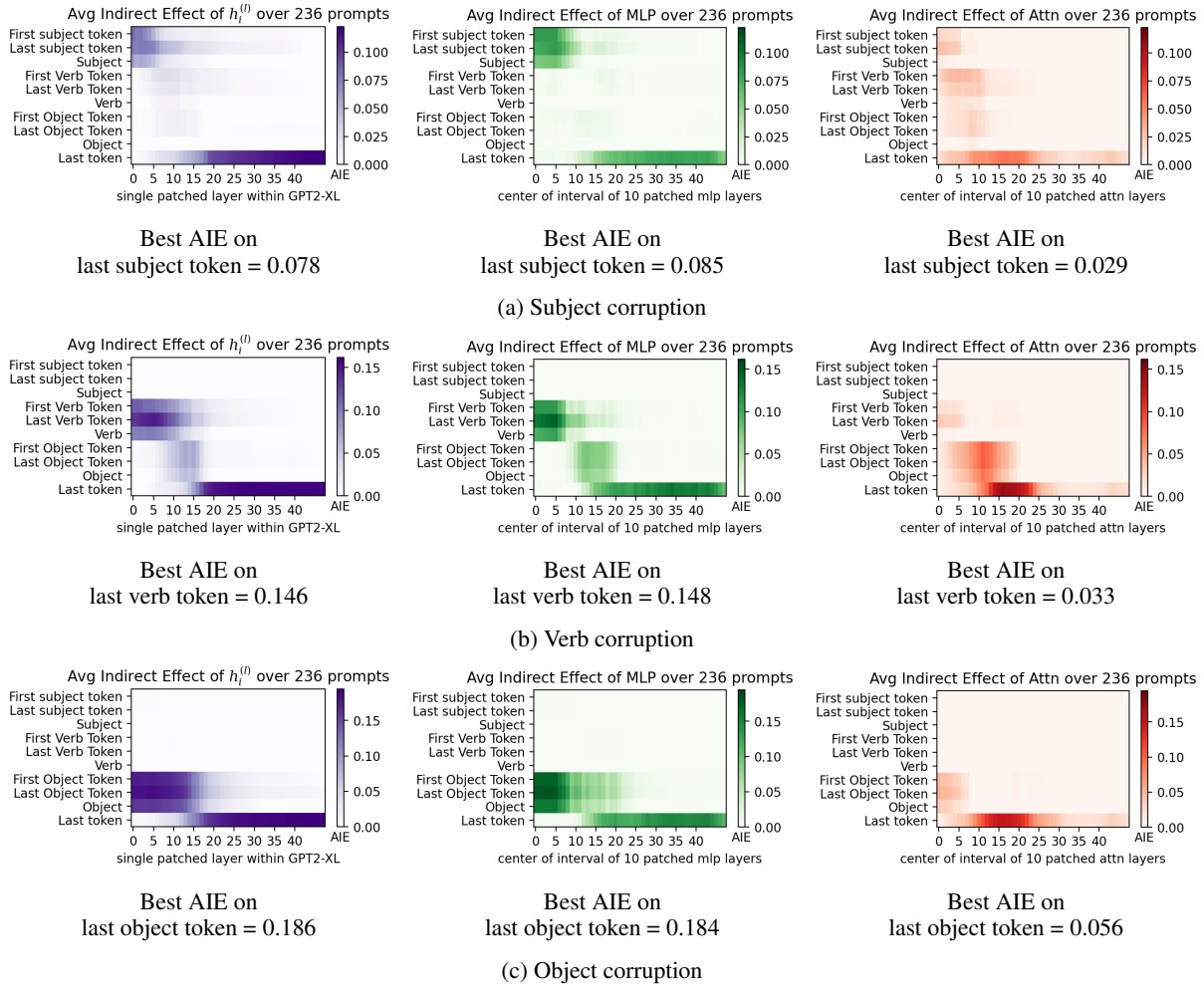

Best AIE on
last subject token = 0.078

Best AIE on
last subject token = 0.085

Best AIE on
last subject token = 0.029

(a) Subject corruption

Best AIE on
last verb token = 0.146

Best AIE on
last verb token = 0.148

Best AIE on
last verb token = 0.033

(b) Verb corruption

Best AIE on
last object token = 0.186

Best AIE on
last object token = 0.184

Best AIE on
last object token = 0.056

(c) Object corruption

Figure 3: Causal tracing for GPT-2 XL Base Model on PEP3k EDIT VALIDATION SET when different tokens are corrupted, $\{s, v, o\}$ (in order). See Appendix A.10 for GPT-2 Large and 20Q results.

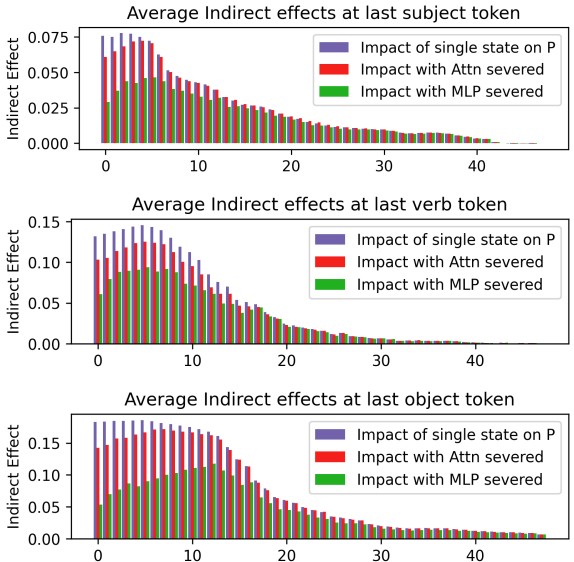

Figure 4: Severed causal tracing results for $\{s, v, o\}$ for GPT-2 XL base on PEP3k EDIT VALIDATION SET

| Dataset | Model | MEMIT F1 Score % | MEMIT_CSK F1 Score % |
|---------|-------|------------------|----------------------|
| PEP3K | GPT-2 Large | 88.53 | 93.78 (+5.25) |
| | GPT-2 XL | 90.51 | 95.09 (+4.58) |
| 20Q | GPT-2 Large | 85.31 | 87.09 (+1.78) |
| | GPT-2 XL | 90.32 | 92.31 (+1.99) |

Table 3: Comparison of MEMIT and best performing MEMIT_CSK on EDIT VALIDATION SET. MEMIT editing is on $s$, while MEMIT_CSK is on best among $\{s, v, o\}$.

## 5.3 MEMIT_CSK exhibits configuration generalization

Table 4 reports GPT-2 XL results for EDIT VALIDATION SET and EDIT SET [9]. The GPT-2 Large results are in Appendix A.6 Table 12. For the EDIT VALIDATION SET performance, the *verb* edit F1 score is higher by **+17.97%** compared to the Base

---

[9]The best hyperparameters are detailed in Appendix A.5. The KL divergence and cut-off factors can potentially enhance the performance for editing methods, see Appendix A.8.

| Dataset | Update Method | Edit Token | Edit Layers | EDIT VALIDATION SET | | | EDIT SET | | |
|---|---|---|---|---|---|---|---|---|---|
| | | | | F1 Score % | Efficacy % | Relapse % | F1 Score % | Efficacy % | Relapse % |
| PEP3k | Base Model | - | - | 77.12 | 0 | 0 | 76.47 | 0 | 0 |
| | RFT$_{\text{Early Stop}}$ | - | - | 90.16 (+13.05) | 97.14 | 11.87 | 80.93 (+4.46) | 50.83 | 9.82 |
| | RFT$_{\text{Fixed Epoch}}$ | - | - | 90.16 (+13.05) | 97.14 | 11.87 | 56.89 (-19.58) | 98.89 | 55.25 |
| | Edit | Last Subject | 1,2,3,4,5 | 90.51 (+13.39) | 80 | 6.36 | 84.72 (+8.25) | 77.22 | 12.98 |
| | Edit | Last Verb | 6,7,8 | **95.09 (+17.97)** | **92.86** | **4.24** | **91.90 (+15.43)** | **88.33** | **7.00** |
| | Edit | Last Object | 3,4,5 | 94.43 (+17.32) | 91.43 | 4.66 | 86.69 (+10.22) | 72.78 | 8.97 |
| 20Q | Base Model | - | - | 74.73 | 0 | 0 | 75.77 | 0 | 0 |
| | RFT$_{\text{Early Stop}}$ | - | - | 85.71 (+10.98) | 80.46 | 12.40 | 77.36 (+1.60) | 30.97 | 7.8 |
| | RFT$_{\text{Fixed Epoch}}$ | - | - | 85.71 (+10.98) | 80.46 | 12.40 | 48.02 (-27.74) | 88.63 | 64.96 |
| | Edit | Last Subject | 2,3,4,5,6 | **92.31 (+17.58)** | 79.69 | **3.43** | 86.46 (+10.70) | 65.73 | **6.90** |
| | Edit | Last Verb | 3,4,5,6,7 | 82.64 (+7.91) | 44.53 | 4.49 | 79.03 (+3.27) | 35.91 | 7.11 |
| | Edit | Last Object | 1,2,3 | 91.12 (+16.39) | **89.06** | 8.18 | **88.09 (+12.33)** | **76.60** | 8.21 |

Table 4: Configuration generalization results based on the best hyperparameters identified for EDIT VALIDATION SET and applied to EDIT SET for GPT-2 XL. The editing methods display high configuration generalization compared to repair-finetuning. Refer to §5.3 for further discussion. GPT-2 Large results are in Appendix A.6 Table 12.

Model in PEP3k. The *object* edit F1 score is higher by **+17.58%** in 20Q. This indicates the importance of varying editing tokens. The best editing method outperforms repair-finetuning baseline consistently for both datasets with much lower relapsed scores.

The editing method *continues to perform well* after transferring the best hyperparameters to EDIT SET; in comparison, both repair-finetuning baselines performance drops significantly. Noticeably, RFT$_{\text{Fixed Epoch}}$ method has high efficacy but a much higher relapse score, between 38.36-64.96%, causing a significant decrease in the F1 score due to overfitting. The three editing methods on $\{s, v, o\}$ outperform the repair-finetuning methods by **10.54-15.43%** for the updated F1 score, exhibiting a better configuration generalization performance.

### 5.4 MEMIT$_{\text{CSK}}$ exhibits semantic generalization

Table 5 shows GPT-2 XL results on PROBE SET[10]. Compared to the editing methods, the repair-finetuning baselines struggle to balance the affected and unaffected samples. RFT$_{\text{Early Stop}}$ performs well in unaffected neighborhoods but struggles with the affected statements (measured by average). RFT$_{\text{Fixed Epoch}}$ reached higher performance on affected subsets but suffered with unaffected neighborhoods. In comparison, the editing methods showed balanced improvements across metrics.

We also noticed that the affected neighborhood scores are generally high except for the specific editing token; e.g., while editing the object token, the affected object neighborhood score is low.

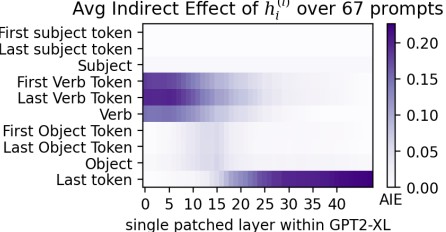

Verb corruption, best AIE on last verb token = 0.200

(a) RFT$_{\text{Early Stop}}$ model with 90.16% F1 Score

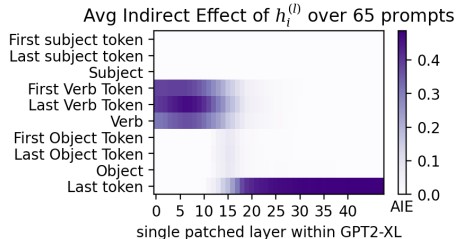

Verb corruption, **best AIE on last verb token = 0.468**

(b) MEMIT$_{\text{CSK}}$ $v$ edited model with 95.09% F1 Score

Figure 5: Causal tracing for GPT-2 XL models on successfully **corrected** statements in the PEP3k EDIT VALIDATION SET. For the RFT$_{\text{Early Stop}}$ model, we observe similar patterns as Fig. 3 for both token corruptions. For the edited model, an improved pattern is observed at $v$.

### 5.5 MEMIT$_{\text{CSK}}$ outperforms fine-tuning for repairing commonsense knowledge

To measure improvement, we re-conduct causal analysis via each token $\{s, v, o\}$ corruption using *successfully* edited statements. Fig. 5 displays the causal graphs for best-performing edit: $v$ edited model and the best repair-finetuned model: RFT$_{\text{Early Stop}}$ based on Table 4.

For RFT$_{\text{Early Stop}}$ (F1 Score 90.16%), the overall

---

[10]Base Model has 0% efficacy on PROBE SET by design.

| Dataset | Update Method | Edit Token | Efficacy % | Unaffected Neighborhood % | | Affected Neighborhood % | | | Affected Paraphrase % | Affected Reasoning % | Average Unaffected % | Average Affected % |
|---|---|---|---|---|---|---|---|---|---|---|---|---|
| | | | | Subject | Object | Subject | Verb | Object | | | | |
| **PEP3k** | Base Model | - | 0 | 100 | 100 | 21.01 | 23.45 | 23.3 | 33.64 | 31.13 | 100 | 26.51 |
| | RFT$_{Early\ Stop}$ | - | 39.63 | 77.21 | 76.98 | 33.88 | 37.5 | 39.16 | 39.91 | 55.66 | 77.09 | 41.22 |
| | RFT$_{Fixed\ Epoch}$ | - | 99.62 | 24 | 27.78 | 86.35 | 84.31 | 87.15 | 68.7 | 62.26 | 25.89 | 77.75 |
| | Edit | Last Subject | 72.08 | **81.21** | **64.15** | 27.75 | **56.83** | 58.67 | **47.92** | **35.85** | **72.68** | 45.40 |
| | Edit | Last Verb | **87.92** | 59.24 | 57.06 | 57.36 | 34.47 | **71.05** | 37.41 | 31.13 | 58.15 | **46.28** |
| | Edit | Last Object | 75.47 | 58.11 | 57.82 | **57.52** | 54.35 | 27.86 | 43.99 | 34.72 | 57.96 | 43.69 |
| **20Q** | Base Model | - | 0 | 100 | 100 | 33.02 | 24.78 | 29.38 | 33.70 | 39.38 | 100 | 32.05 |
| | RFT$_{Early\ Stop}$ | - | 21.52 | 87.01 | 86.89 | 36.21 | 34.72 | 37.01 | 38.27 | 35.41 | 86.95 | 36.32 |
| | RFT$_{Fixed\ Epoch}$ | - | 79.27 | 38.85 | 37.89 | 67.40 | 64.41 | 64.44 | 60.20 | 40.72 | 38.37 | 59.43 |
| | Edit | Last Subject | 61.94 | 87.22 | 71.00 | 34.48 | 57.47 | **58.82** | **51.75** | 37.93 | 79.11 | 48.09 |
| | Edit | Last Verb | 35.70 | **90.18** | 83.21 | 37.44 | 29.80 | 47.25 | 35.91 | 37.80 | **86.69** | 37.64 |
| | Edit | Last Object | **72.18** | 76.24 | **92.26** | **61.90** | 60.37 | 33.76 | 47.87 | **41.90** | 84.25 | **49.16** |

Table 5: Efficacy and semantic generalization results on PROBE SET for GPT-2 XL. Balanced improvements are observed for editing methods across metrics, with the $s$ and $o$ edits performing the best. Refer to §5.4 for a detailed discussion. GPT-2 Large results are in Appendix A.6 Table 13.

causal pattern and AIE remain similar to the Base Model in Fig. 3. In contrast, the $v$ edited model (F1 Score 95.09%) shows an enhanced AIE for all types of corruption. Specifically, a **high AIE of 0.468** is recorded at the last verb token for verb corruption. These findings confirm that localization and AIE improve for the edited model at the edit location.

# 6 Related Work

Early works on model editing focused on updating individual neurons using constrained finetuning (Sinitsin et al., 2020; Zhu et al., 2020) or hypernetworks (De Cao et al., 2021; Mitchell et al., 2022a; Hase et al., 2023b). A related line of work has focused on storing updates in an external memory (Jin et al., 2021; Mitchell et al., 2022b; Tandon et al., 2022, *inter alia*). Recent works (Hoelscher-Obermaier et al., 2023; Zhong et al., 2023; Brown et al., 2023; Onoe et al., 2023) offer more comprehensive evaluations for fact-editing methods.

Inspired by the linear associative memory property of feedforward layers in Transformers (Anderson, 1972; Geva et al., 2021, 2022) and success with the approach in convolutional models (Bau et al., 2020), recent works have proposed to edit MLP weights directly (Meng et al., 2022; Dai et al., 2022; Yao et al., 2022). In the encyclopedic factual domain, Meng et al. (2022) proposed to edit single facts by fitting a Rank One Model Edit (ROME) to the parameters of an MLP layer, and showed it outperformed prior methods. Our work builds on Meng et al. (2023), which extended this approach to thousands of edits by altering the weights of a range of MLP layers. Hase et al. (2023a) demonstrate that many early edit layers can work well

with MEMIT; this partially motivates our extensive layer hyperparameter search. Recent work by Cohen et al. (2023) proposes a dataset for evaluation of a variety of ripple effects in editing methods with factual knowledge and concludes that models fail to capture these effects. All aforementioned works focus on encyclopedic factual knowledge, unlike ours.

# 7 Conclusion

This paper demonstrates strong causal relations between commonsense plausibility judgments and early MLP layers in Transformers. These parameters are directly editable for repairing commonsense mistakes. We improve the MEMIT parameter editing algorithm to MEMIT$_{CSK}$ for commonsense plausibility prediction by varying edit tokens and by improving the layer selection strategy. GPT-2 Large and XL models edited by MEMIT$_{CSK}$ outperform repair-finetuned baselines by more than 10% F1 score on EDIT SET. Additionally, we construct a PROBE SET that contains unaffected and affected neighborhoods, affected paraphrases, and affected reasoning challenges for comprehensive evaluation. MEMIT$_{CSK}$ effectively generalizes on related and unrelated neighborhoods annotated in our PROBE SET, exhibiting semantic generalization while repair-finetuned baselines demonstrate significant trade-offs between unaffected and affected metrics. These results indicate a compelling direction of incorporating feedback about common sense in transformers on the fly through direct model editing.

## Limitations

In this work, we experiment with repairing commonsense mistakes by the GPT-2 Large and XL models. We are unable to investigate larger open-sourced models like GPT-J (Wang and Komatsuzaki, 2021) and GPT-NeoX (Black et al., 2022) due to resource limitations. Investigating the research questions described in §3 on larger models is a natural next step. We focus on the binary plausibility prediction task but envision that parameter editing could improve models on various commonsense tasks in future work.

Our experiments show that the optimal edit token (subject, verb, or object) varies among datasets. The specific location of a single generalized optimal edit token, if it exists, requires further investigation, while different editing methods for commonsense knowledge can be proposed.

## Ethics Statement

This study proposes a framework to evaluate and correct commonsense mistakes in GPT-2 models, focusing on predicting the plausibility of commonsense statements. Commonsense knowledge is highly contextualized and varies significantly across locations and cultures. Biases and stereotypes present in edit datasets may inadvertently lead to erroneous and potentially harmful model judgments. Malicious actors may exploit model editing to incorporate false information into models. It is crucial to employ meticulously curated datasets in future research and during the deployment of these models in real-world scenarios.

## Acknowledgments

This research was conducted at the University of Massachusetts Amherst under the Industry Mentorship Program led by Prof. Andrew McCallum. We are grateful for their support and resources provided for this research.

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

# A  Appendix

## A.1  Causal Tracing Background

Given a model, the method takes a concatenation of subject $s$ and verb $v$ as input prompt $x$, then predicts the corresponding object $o$ as prediction $y$. For example, for the statement "*Paris is the capital of*", a model is tasked with predicting "*France*" as the most-likely next token. Taking a correctly predicted $x, y$ pair, Causal tracing consists of the following three steps:

**Step 1: clean run.** Given the input prompt $x$, they collect all hidden activation values $\left\{ h_i^l \mid i \in [1, T], l \in [1, L] \right\}$ from the model, where $T$ is number of input tokens in $x$ and $L$ is number of model layers. Concretely, for each input $x$, $h_i^l(x) = h_i^{l-1}(x) + a_i^l(x) + m_i^l(x)$ where $a_i^l$ is the attention value and $m_i^l$ is the corresponding MLP value. The predicted probability of the correct object is denoted as $\mathbb{P}[y]$.

**Step 2: corrupted run.** In this setting, certain part of the input prompt $x$ is corrupted with noise. In a clean run, $x$ is embedded as $\left[ h_1^{(0)}, h_2^{(0)} \dots h_T^{(0)} \right]$. However, here, they set $h_i^{(0)} := h_i^{(0)} + \epsilon$, for all tokens $i$ in the subject token[11]. The probability of ground truth value $y$ produced in this run is denoted as $\mathbb{P}_*[y]$. Note that the model prediction is likely to be incorrect due to the noisy input.

---

[11] $\epsilon \sim \mathcal{N}(0, \ v)$ and $v$ is taken as three times the empirical standard deviation of the embeddings corresponding to the subject tokens.

| Dataset | $N_{\text{Train}}$ | $N_{\text{EV}}$ | $N_{\text{E}}$ | $N_{\text{P}}$ |
|---------|---------|---------|--------|--------|
| **PEP3k** | 1,225 | 306 | 1,531 | 265 |
| **20Q** | 2,006 | 507 | 2,548 | 381 |

Table 6: Number of samples in the Training Set, EDIT VALIDATION SET, EDIT SET, and PROBE SET.

| Type | $N_{\text{PEP3k}}$ | $N_{\text{20Q}}$ |
|------|---------|--------|
| Original statement | 265 | 381 |
| Unaffected subject neighborhood | 1,325 | 1,894 |
| Unaffected object neighborhood | 1,325 | 1,900 |
| Affected subject neighborhood | 1,290 | 1,856 |
| Affected verb neighborhood | 1,288 | 1,832 |
| Affected object neighborhood | 1,292 | 1,848 |
| Affected paraphrase | 1,323 | 1,905 |
| Affected reasoning | 530 | 754 |

Table 7: Number of samples in the PROBE SET.

**Step 3: corrupted-with-restoration-run.** The model runs inference using the noisy input embedding created in the corrupted run, with the difference that the model is also forced to output the clean state activation $h_{\hat{i}}^{\hat{l}}$ at certain token $\hat{i}$ and layer $\hat{l}$. If the model successfully produces the correct output using a small number of clean states, there is likely to be a strong casual relationship between these states and the model output. The probability of the correct object is denoted as $\mathbb{P}_{*,\text{clean }h_i^{(l)}}[y]$.

The three runs produced $\mathbb{P}[y]$, $\mathbb{P}_*[y]$ and $\mathbb{P}_{*,\text{clean }h_i^l}[y]$. Two metrics are then defined to measure the states effect between these runs. **Total effect (TE)** is calculated as $\mathbb{P}[y] - \mathbb{P}_*[y]$, while the **indirect effect (IE)** of a specific hidden state $h_i^l$ is calculated as $\mathbb{P}_{*,\text{clean }h_i^{(l)}}[y] - \mathbb{P}_*[y]$. The average total effect, **ATE** and average indirect effect, i.e. **AIE**, are computed across multiple examples for each hidden state.

## A.2 Datasets

**Physical Event Plausibility** (PEP3k; Wang et al., 2018) consists of 3,062 statements in (subject *s*, verb *v*, object *o*) format about semantically plausible and implausible events. It covers a wide range of possible (but not necessarily common) events with high annotator label agreement.

**20 Questions** (20Q)[12] is a dataset of 5,096 commonsense statements written by crowd annotators

[12] https://github.com/allenai/twentyquestions

in games of "20 questions" and labeled as plausible or implausible. We use the (s,v,o) format of the dataset constructed by Porada et al. (2021), where $x = (s, v, o)$ and $y \in \{True, False\}$.

Examples from each dataset are given in Table 1. Statistics of our created data splits are in Tables 6 and 7

## A.3 Base Model vs. Zero-Shot for 20Q Dataset

Comparison of Base Model and zero-shot model for the 20Q dataset is in Fig. 6.

## A.4 Original MEMIT Editing Results

Table 8 shows the detailed metrics and editing parameters for MEMIT applied on EDIT VALIDATION SET.

| Dataset | Model | Edit Token | Edit Layers | F1 Updated % | Efficacy % | Relapse % |
|---------|-------|------------|-------------|--------------|------------|-----------|
| PEP3K | GPT-2 Large | Subject | 4,5,6,7,8 | 88.53 (+13.36) | 76.32 | 7.39 |
| | GPT-2 XL | Subject | 1,2,3,4,5 | 90.51 (+13.39) | 80 | 6,36 |
| 20Q | GPT-2 Large | Subject | 1,2,3,4,5 | 85.31 (+12.92) | 71.43 | 9.26 |
| | GPT-2 XL | Subject | 1,2,3 | 90.32 (+15.59) | 84.38 | 7.65 |

Table 8: Editing results after applying original MEMIT on EDIT VALIDATION SET.

## A.5 Hyperparameters

**Base Finetuning** The GPT-2 Large and XL models are initially finetuned on the training set with the next-token prediction objective. Table 9 presents the optimal hyperparameters identified for the base-finetuning method.

| Dataset | Model | Learning Rate | Batch Size | Epochs |
|---------|-------|---------------|------------|--------|
| 20q | GPT-2 Large | 0.00009961 | 64 | 10 |
| | GPT-2 XL | 0.00001432 | 64 | 10 |
| PEP3k | GPT-2 Large | 0.00002298 | 8 | 10 |
| | GPT-2 XL | 0.00001023 | 32 | 20 |

Table 9: Base Model hyperparameters for Training Set

**Repair Finetuning**

Table 10 shows the best hyperparameters for the repair-finetuning method. The method was very sensitive to small changes in learning rate while the other parameters worked well over a long range of values. Note that we use early stopping and restore the weights to the best performing model based on the F1 score.

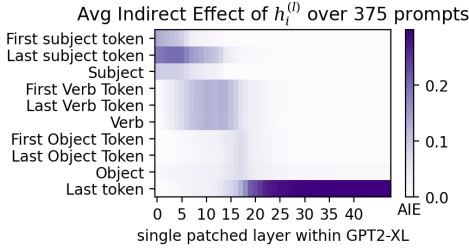 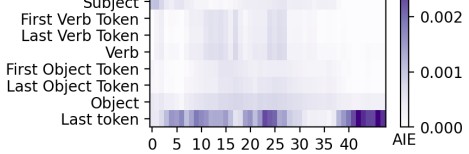

(a) Base Model with 73.96% accuracy.      (b) Zero-shot model with 51.87% accuracy.

Figure 6: Zero-shot vs. Base Model causal tracing results for GPT-2 XL on 20Q EDIT VALIDATION SET.

| Dataset | Model | Learning Rate | Batch Size | Epochs |
|---|---|---|---|---|
| 20q | GPT-2 Large | 0.000003451 | 8 | 7 |
| | GPT-2 XL | 0.000001589 | 32 | 9 |
| PEP3k | GPT-2 Large | 0.00000474 | 32 | 7 |
| | GPT-2 XL | 0.000001313 | 8 | 10 |

Table 10: Hyper-parameters for RFT$_{\text{Fixed Epoch}}$ tuned for EDIT VALIDATION SET and applied to EDIT SET and PROBE SET

**MEMIT$_{\text{CSK}}$**

The Table 11 shows the hyper-parameters for the editing method. The method was slightly sensitive to the learning rate and very sensitive to the edit token. Note that a KL divergence factor of 0.0625 was used as the default value for all editing experiments. Appendix A.8 contains an ablation study of the KL divergence factor.

| Dataset | Model | Edit Token | Layers | Learning Rate |
|---|---|---|---|---|
| 20q | GPT-2 Large | Last Subject | 3,4,5 | 0.7868 |
| | | Last Verb | 2,3,4,5,6 | 0.09393 |
| | | Last Object | 1,2,3 | 0.6276 |
| | GPT-2 XL | Last Subject | 2,3,4,5,6 | 0.04108 |
| | | Last Verb | 3,4,5,6,7 | 0.01936 |
| | | Last Object | 1,2,3 | 0.02689 |
| PEP3k | GPT-2 Large | Last Subject | 4,5,6,7,8 | 0.32 |
| | | Last Verb | 4,5,6,7,8 | 0.682 |
| | | Last Object | 1,2,3,4,5 | 0.433 |
| | GPT-2 XL | Last Subject | 1,2,3,4,5 | 0.1253 |
| | | Last Verb | 6,7,8 | 0.08719 |
| | | Last Object | 3,4,5 | 0.04107 |

Table 11: Hyper-parameters for the editing method tuned for EDIT VALIDATION SET and applied to EDIT SET and PROBE SET.

### A.6 GPT-2 Large Results for Configuration and Semantic Generalization

The GPT-2 Large results for configuration generalization experiments are in Table 12. The GPT-2 Large results for semantic generalization experiments are in Table 13.

### A.7 Layer Selection Strategy

For demonstration purposes let's assume our model has only 10 layers. The average indirect effects of these layers at our desired edit token (let's assume last verb token) are:

$$[0.0, 0.1, 0.2, 0.3, 0.5, 0.4, 0.4, 0.3, 0.2, 0, 0]$$

Let's also assume that we are considering only 5 layer windows. The highest average indirect effect is observed at the 5th layer with value 0.5. According to MEMIT, the optimal edit layers will be a 5 layer window ending at the highest AIE layer, in this case it will be the layers $1, 2, 3, 4, 5$.

Now let's calculate the moving average of 5 layer windows. The moving average of layers 1-5 is $(0.0 + 0.1 + 0.2 + 0.3 + 0.5)/5 = 0.22$, similarly the moving average of layers 2-6 will be $(0.1 + 0.2 + 0.3 + 0.5 + 0.4)/5 = 0.3$ and so on. The moving averages of all 5 layer windows are:

$$[0.22, 0.3, 0.36, 0.38, 0.36, 0.26]$$

The maximum moving average is observed for layers 4-8 with value 0.38. In our method, we would also consider layers 4-8 as in our hyperparameter search space along with layers 1-5.

### A.8 Ablation Study

**KL Divergence Factor**

The Table 14 shows how the performance of the editing method changes when varying the KL Divergence Factor in terms of Accuracy and F1 score. The ablation study is conducted using the GPT-2 Large model on the PEP3k dataset, and the verb token is used for editing in the EDIT VALIDATION SET dataset. The chosen hyperparameters align with those presented in Table 11.

| Dataset | Update Method | Edit Token | Edit Layers | EDIT VALIDATION SET | | | EDIT SET | | |
|---|---|---|---|---|---|---|---|---|---|
| | | | | F1 Score % | Efficacy % | Relapse % | F1 Score % | Efficacy % | Relapse % |
| PEP3k | Base Model | - | - | 75.16 | 0 | 0 | 76.22 | 0 | 0 |
| | RFT$_{\text{Early Stop}}$ | - | - | 95.75 (+20.59) | 94.74 | 3.91 | 80.92 (+4.70) | 40.93 | 6.60 |
| | RFT$_{\text{Fixed Epoch}}$ | - | - | 95.75 (+20.59) | 94.74 | 3.91 | 51.08 (-19.14) | 100 | 55.70 |
| | Edit | Last Subject | 4,5,6,7,8 | 88.53 (+13.36) | 76.32 | 7.39 | 79.36 (+3.14) | 54.95 | 12.77 |
| | Edit | Last Verb | 4,5,6,7,8 | 93.78 (+18.62) | **96.05** | 6.96 | 89.08 (+12.86) | **93.68** | **12.34** |
| | Edit | Last Object | 1,2,3,4,5 | 88.41 (+13.25) | 86.84 | 10.87 | 77.65 (+1.43) | 78.57 | 21.85 |
| 20Q | Base Model | - | - | 72.39 | 0 | 0 | 74.07 | 0 | 0 |
| | RFT$_{\text{Early Stop}}$ | - | - | 91.32 (+18.93) | 97.86 | 11.17 | 76.45 (+2.37) | 48.23 | 13.69 |
| | RFT$_{\text{Fixed Epoch}}$ | - | - | 91.32 (+18.93) | 97.86 | 11.17 | 69.92 (-4.15) | 94.61 | 38.36 |
| | Edit | Last Subject | 3,4,5 | 85.33 (+12.94) | 75 | 10.63 | 81.97 (+7.90) | 67.18 | 12.66 |
| | Edit | Last Verb | 2,3,4,5,6 | 77.64 (+5.25) | 38.57 | **7.36** | 77.33 (+3.26) | 33.44 | **7.22** |
| | Edit | Last Object | 1,2,3 | **87.09 (+14.71)** | **82.14** | 10.90 | **84.61 (+10.54)** | **80.43** | 13.79 |

Table 12: Configuration generalization results based on the best hyperparameters identified for the EDIT VALIDATION SET and applied to the EDIT SET for GPT-2 Large. The editing method displays high configuration generalization while both variants of the repair-finetuning method have a lower F1 Score on the EDIT SET. Refer to §5.3 for further discussion.

| Dataset | Update Method | Edit Token | Efficacy % | Unaffected Neighborhood % | | Affected Neighborhood % | | | Affected Paraphrase % | Affected Reasoning % | Average Unaffected % | Average Affected % |
|---|---|---|---|---|---|---|---|---|---|---|---|---|
| | | | | Subject | Object | Subject | Verb | Object | | | | |
| PEP3k | Base Model | - | 0 | 100 | 100 | 19.77 | 18.94 | 23.84 | 34.16 | 32.08 | 100 | 25.76 |
| | RFT$_{\text{Early Stop}}$ | - | 30.57 | 83.92 | 82.64 | 30.93 | 30.67 | 35.53 | 38.85 | 33.77 | 83.28 | 33.95 |
| | RFT$_{\text{Fixed Epoch}}$ | - | 100.00 | 19.47 | 26.79 | 93.02 | 92.86 | 92.03 | 80.73 | 36.23 | 23.13 | 78.97 |
| | Edit | Last Subject | 58.87 | **79.01** | 69.51 | 27.05 | 50.70 | 54.26 | 45.73 | 40.57 | **74.26** | 43.66 |
| | Edit | Last Verb | **96.23** | 44.15 | 48.06 | 69.92 | 38.28 | **83.82** | 41.65 | 33.02 | 53.34 | 53.34 |
| | Edit | Last Object | 82.26 | 44.15 | **73.28** | **71.78** | 69.25 | 36.15 | **53.74** | **46.04** | 58.71 | **55.3** |
| 20Q | Base Model | - | 0 | 100 | 100 | 30.23 | 22.93 | 27.11 | 32.76 | 27.72 | 100 | 28.15 |
| | RFT$_{\text{Early Stop}}$ | - | 29.66 | 88.07 | 87.05 | 39.17 | 37.34 | 39.23 | 42.05 | 27.59 | 87.56 | 37.08 |
| | RFT$_{\text{Fixed Epoch}}$ | - | 95.01 | 55.91 | 45 | 71.22 | 77.07 | 70.07 | 66.29 | 30.10 | 50.45 | 62.95 |
| | Edit | Last Subject | 67.98 | 79.57 | 57.79 | 35.08 | 63.26 | **63.91** | **53.96** | 31.70 | 68.68 | 49.58 |
| | Edit | Last Verb | 32.55 | **89.55** | 84.16 | 37.93 | 26.80 | 46.10 | 35.17 | 28.51 | **86.85** | 34.90 |
| | Edit | Last Object | **81.89** | 66.95 | **85.32** | **71.98** | 71.67 | 34.79 | 51.60 | **35.68** | 76.13 | **53.14** |

Table 13: Efficacy and semantic generalization results for the PROBE SET for GPT-2 Large. Balanced improvements are observed for editing methods across metrics, with the object token editing method performing the best. In comparison, the repair-finetuning models show skewed performance between unaffected and affected metrics. Refer to §5.4 for a detailed discussion.

## Cut-Off Factor

This hyperparameter is introduced to "early stop" the optimization step.[13] When the probability of $y_i$ exceeds this cut-off factor upon adding the residual $\delta_i$ to the transformer's hidden state $h_i^L$, the optimization step is stopped.

The Table 15 demonstrates how the performance of the editing method changes when varying the "Cut-Off" Factor in terms of Accuracy and F1 score. The ablation study is conducted using the GPT-2 Large model on the PEP3k dataset, with the verb token used for editing in the EDIT VALIDATION SET dataset. The chosen hyperparameters align with those presented in Table 11.

---

[13]Please refer to (Meng et al., 2023) for details of the optimization equation.

## A.9 Constructing the PROBE SET

We prompt text-davinci-003 zero-shot to construct the augmentations for each test instance; the prompts are given in:

- Affected Paraphrase: Fig. 7

- Affected Reasoning: Fig. 8

- Affected Neighborhood: Fig. 9

- Unaffected Neighborhood: Fig. 10

We prompt the model for 5 possible instances, but it can sometimes return the same value multiple times. We filter out poorly-formatted instances and manually clean the filtered data to remove things like empty statements or incorrect parsing from

| KL Div. Factor | Efficacy | Accuracy | F1 score |
|---|---|---|---|
| Base M | - | 75.16 | 75.16 |
| 0.001 | 88.16 | 91.83 | 91.81 |
| 0.0025 | 88.16 | 91.83 | 91.81 |
| 0.005 | 89.47 | 92.16 | 92.14 |
| 0.0075 | 89.47 | 92.16 | 92.14 |
| 0.01 | 89.47 | 92.16 | 82.06 |
| 0.025 | 86.84 | 91.17 | 91.15 |
| 0.05 | 92.10 | 92.16 | 92.13 |
| 0.0625 | 90.79 | 91.83 | 91.81 |
| 0.075 | **93.42** | **92.81** | **92.80** |
| 0.1 | 92.11 | 91.83 | 91.81 |
| 0.25 | 93.42 | 91.18 | 91.14 |
| 0.5 | 92.11 | 90.85 | 90.83 |
| 0.75 | 93.42 | 91.5 | 91.48 |
| 1 | 92.11 | 91.18 | 91.16 |

Table 14: Ablation study of the KL Divergence Factor on the GPT-2 Large model edited using the verb token on layers $l \in 2, 3, 4, 5, 6$ in the EDIT VALIDATION SET split of PEP3k. Note that default KL Factor of 0.0625 is used to report the performance of all editing methods.

| Cut-Off Factor | Efficacy | Accuracy | F1 score |
|---|---|---|---|
| Base M | - | 75.16 | 75.16 |
| 0.7 | 53.95 | 83.33 | 82.97 |
| 0.725 | 55.26 | 83.33 | 82.97 |
| 0.75 | 55.26 | 83.00 | 82.61 |
| 0.775 | 55.26 | 83.00 | 82.61 |
| 0.8 | 56.58 | 83.66 | 83.35 |
| 0.825 | 63.16 | 84.97 | 84.73 |
| 0.85 | 77.63 | 89.87 | 89.80 |
| 0.875 | 82.90 | 91.50 | 91.47 |
| 0.9 | 90.79 | 92.81 | 92.80 |
| 0.925 | 90.79 | **93.14** | **93.13** |
| 0.95 | 88.16 | 91.50 | 91.48 |
| No factor | 90.79 | 91.83 | 91.81 |

Table 15: Ablation study of the "Cut-Off" Factor on the GPT-2 Large model edited using the verb token on layers $l \in 2, 3, 4, 5, 6$ for PEP3k EDIT VALIDATION SET. Note that the default value of "No Factor" is used to report the performance of all editing methods, i.e., there was no "early stopping" of the optimization step.

```
Provide 5 paraphrases of: Furnishings make noise

------------------------------------------------
1. Furniture can be noisy.
2. Furniture can create sound.
3. Furniture can produce noise.
4. Furniture can be a source of sound.
5. Furniture can be a source of noise.
```

Figure 7: Prompt to generate affected paraphrase for "Furnishings make noise (false)"

```
Furnishings do not make noise. Explain this
with a 2-step reasoning chain of very short,
simple, connected sentences:

------------------------------------------------
1. Furnishings are inanimate objects.
2. Inanimate objects cannot make noise.
```

Figure 8: Prompt to generate affected reasoning neighborhood for "Furnishings make noise (false)"

GPT2-XL Base Model on the 20Q dataset, and the GPT2-Large Base Model on the PEP3k dataset.

For each of the editing locations, we see that the Last Token has higher AIE towards the later layers of the model which is consistent with the results of MEMIT on encyclopedic knowledge. Focusing on the subject, verb, and object tokens, we see that all of them show high AIE in the early layers of the corrupted tokens and that the effect on the corresponding last corrupted token is more pronounced than that of the first corrupted token. This shows that selecting the last subject/verb/object token and the early layers of the model should give good results for the editing method. These patterns are consistent across all the models and datasets.

GPT output to expected key-value pairs. We manually evaluate some examples to ensure quality. In summary, there can be up to 5 augmentations per augmentation type for each instance.

## A.10 Causal Analysis Results

The Figs. 12 to 14 shows the causal graphs for the GPT2-Large Base Model on the 20Q dataset, the

```
Given the text: Furnishings make noise
subject token: Furnishings
object token: noise

Q1. In the text, replace just the subject token
with a different word. The replaced text should
be a valid sentence. The replaced token can be
a hyponym or similar word of the original
subject token. Write up to 5 such variants.

Q2. In the text, replace just the verb token
with a different word. The replaced text should
be a valid sentence. The replaced token can be
a verb that follows or precedes the original
verb token. Write up to 5 such variants.

Q3. In the text, replace just the object token
with a different word. The replaced text should
be a valid sentence. The replaced token can be
a hyponym or similar word of the original
object token. Write up to 5 such variants.

---------------------------------------------
Q1.
1. Appurtenances make noise
2. Fixtures make noise
3. Accoutrements make noise
4. Decorations make noise
5. Adornments make noise

Q2.
1. Furnishings create noise
2. Furnishings emit noise
3. Furnishings generate noise
4. Furnishings produce noise
5. Furnishings yield noise

Q3.
1. Furnishings make sound
2. Furnishings make clamor
3. Furnishings make din
4. Furnishings make racket
5. Furnishings make uproar
```

Figure 9: Prompt to generate affected neighborhood for "Furnishings make noise (false)"

```
Given:
text: Furnishings make noise
subject token: Furnishings
object token: noise

Q1. Replace the subject token with a completely
unrelated word and make a new text. Make 5 such
replacements.

Q2. Replace the object token with a completely
unrelated word and make a new text. Make 5 such
replacements.

---------------------------------------------
1. Replacing the subject token:
a. Cars make noise
b. Animals make noise
c. People make noise
d. Plants make noise
e. Computers make noise

2. Replacing the object token:
a. Furnishings make music
b. Furnishings make laughter
c. Furnishings make light
d. Furnishings make heat
e. Furnishings make color
```

Figure 10: Prompt to generate unaffected neighborhood for "Furnishings make noise (false)"

```
You are given an input sentence. Fix the
grammar and write the grammatical sentence.

input: furnishing make noise
---------------------------------------------
output: furnishings make noise
```

Figure 11: Prompt to fix grammar in a triple "furnishing make noise"

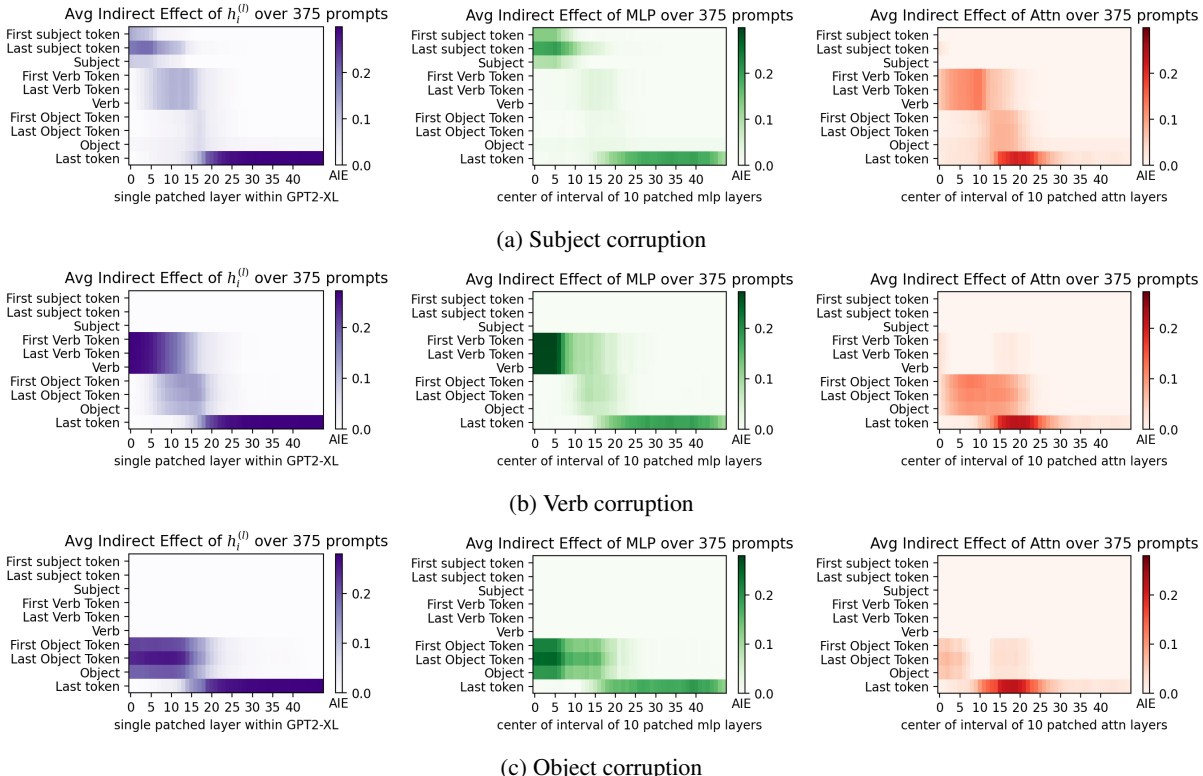

Figure 12: Causal tracing results for GPT-2 XL Base Model on 20Q SMALL CAPS EDIT VALIDATION SET when different parts of the input are corrupted.

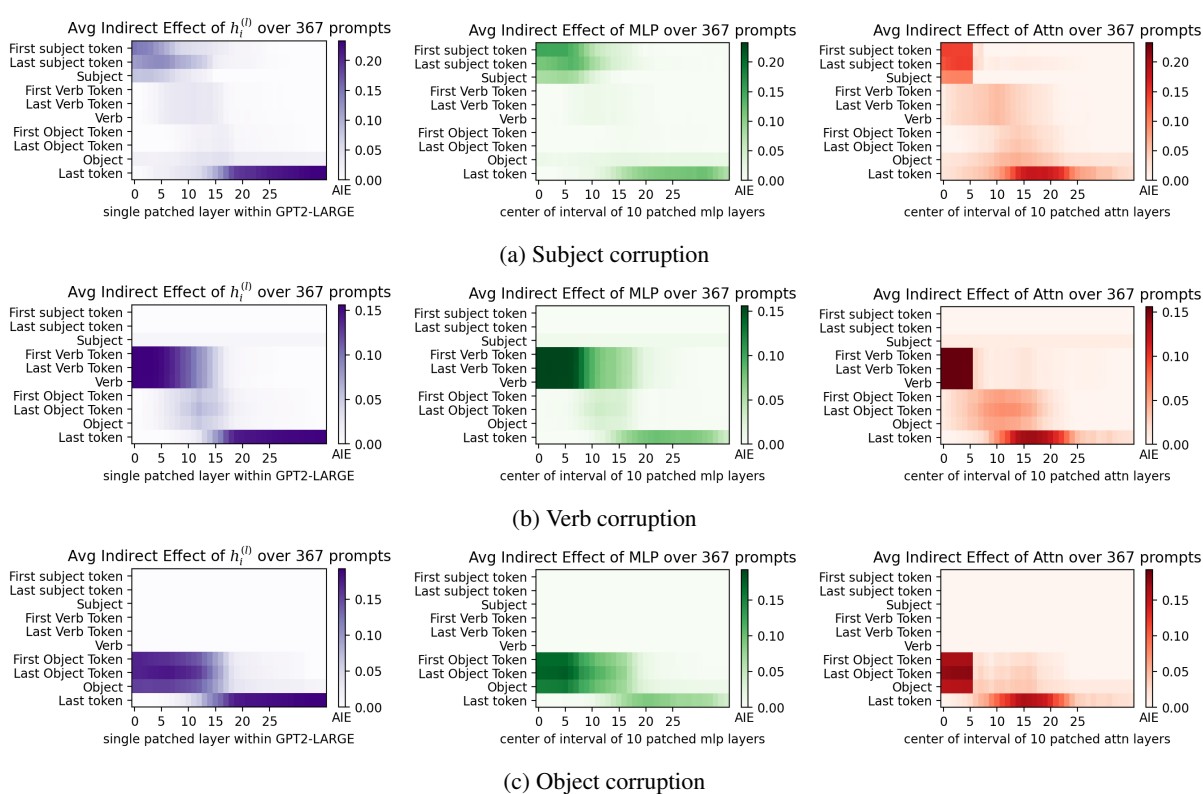

Figure 13: Causal tracing results for GPT-2 Large Base Model on 20Q EDIT VALIDATION SET when different parts of the input are corrupted.

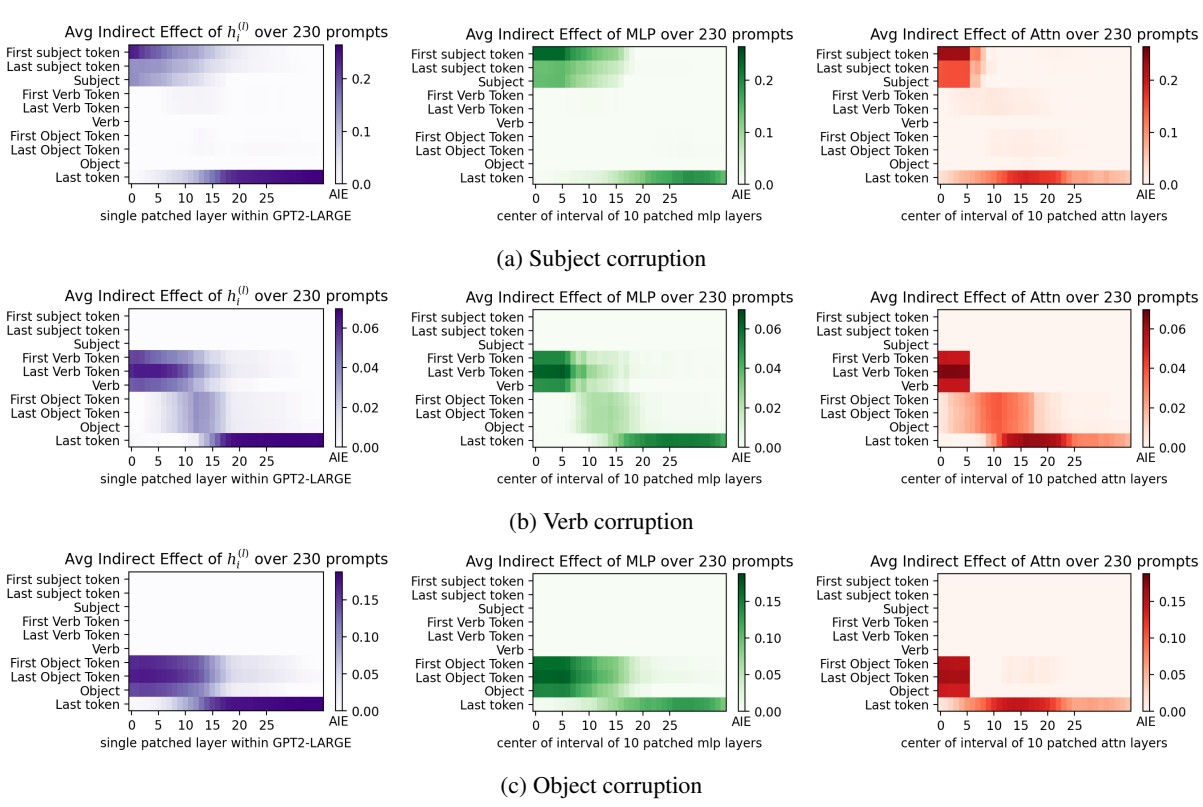

Figure 14: Causal tracing results for GPT-2 Large Base Model on PEP3k EDIT VALIDATION SET when different parts of the input are corrupted.