# OpenReview forum: "Editing Common Sense in Transformers"
_EMNLP/2023/Conference — EMNLP 2023 Main_

### Official Review · Reviewer_4FU4 · 2023-08-02

**Soundness:** 4

**Excitement:**

4: Strong: This paper deepens the understanding of some phenomenon or lowers the barriers to an existing research direction.

**Paper Topic And Main Contributions:**

Topic:

This paper studies whether commonsense knowledge (e.g.,  encyclopedic facts) is causally associated with localized, editable parameters in Transformers, and whether model can repair incorrect commonsense knowledge by updating specific hidden states of Transformers.

Contributions:
1) This paper showcases that there are strong causal relations between commonsense plausibility judgments and early MLP layers in Transformers.
2) This paper improves the MEMIT parameter editing algorithm to MEMIT_{CSK} for commonsense plausibility prediction.
3) A new dataset that contains unaffected and affected neighborhoods, affected paraphrases, and affected reasoning challenges for comprehensive evaluation.

**Reasons To Accept:**

1) Updating language models by editing the model parameters directly (without re-training the model) is a very interesting task. The previous editing methods, however, have only been considered on statements of one-to-one encyclopedic knowledge. The paper first studies the problem for one-to-N commonsense knowledge.
2) The proposed improved version of MEMIT, dubbed MEMIT_CSK, outperforms best fine-tuned baselines by 10%+ on two datasets.
3) A novel dataset for comprehensive evaluation of language model editing.
4) I find the presentation and organization of the paper are generally great. e.g., Fig. 1 provides a clear bigmap of the overall idea.

**Reasons To Reject:**

1) Since the new dataset is also a main contribution. The authors should highlight the differences of the new datasets compared with previous ons, preferably in the last section of intro section where the contributions are highlighted.
2) The authors extend MEMIT editing algorithm, to the commonsense domain, by varying edit tokens and improving the layer selection strategy. The authors highlighted the results of doing so (line 88-103), but the reasons and intuition for doing so are not explained. It is better to explain why doing so before introducing the results.

**Reproducibility:**

4: Could mostly reproduce the results, but there may be some variation because of sample variance or minor variations in their interpretation of the protocol or method.

**Reviewer Confidence:**

2: Willing to defend my evaluation, but it is fairly likely that I missed some details, didn't understand some central points, or can't be sure about the novelty of the work.

---

> ### Author Rebuttal · Authors · 2023-08-29
>
> Thank you for finding our work compelling and valuable, and highlighting that our task is interesting, application area and dataset novel, and presentation and organization strong! We address your concerns below and believe both are easily fixable in the camera-ready.
>
> 1.  **Since the new dataset is also a main contribution. The authors should highlight the differences of the new datasets compared with previous ons, preferably in the last section of intro section where the contributions are highlighted.**
>
> Good point, we will make sure to do this in the camera-ready! Here is a detailed explanation:
>
> The novel dataset we have introduced diverges from prior datasets, such as the ones used in MEND {zsRE question-answering, FEVER fact-checking, Wikitext generation}, and ROME/MEMIT {zsRE question-answering, COUNTERFACTUAL}, in multiple key aspects. Note that, MEND employed various datasets tailored to specific model types {seq2seq (BART/T5), classification (BERT), GPT}, focusing solely on evaluating Efficacy of edit and one task-specific metric like changes in perplexity or accuracy compared to the base model, lacking an assessment of semantic generalization.
>
> Firstly, our dataset was curated using augmentations of instances drawn from the 20 Questions and PEP3K datasets. This dataset presents a nuanced commonsense knowledge, in contrast to the previous datasets which were centered around factual content from encyclopedic sources.
>
> Secondly, our dataset introduces "affected reasoning" statements, to assess if the model correctly comprehends the underlying reasons. As previously highlighted, this inclusion is of paramount importance due to the interconnected nature of commonsense knowledge; modifying one facet of knowledge could potentially influence other associated domains.
>
> Thirdly, our dataset introduces the concept of affected neighborhoods, a dimension that previous datasets lacked. While earlier datasets only encompassed neighborhoods that did not change by model edits (referred to as specificity in ROME/MEMIT or unaffected neighborhoods in our context), our dataset presents both unaffected and affected neighborhoods, contributing to a more comprehensive understanding.
>
> Lastly, during the creation of these affected and unaffected neighborhoods, we augment {S,V,O} and {S,O} respectively. This contrasts with prior datasets that solely accounted for subject changes in unaffected neighborhoods, neglecting considerations for affected neighborhoods, as elaborated earlier.
>
> We will incorporate this explanation in the next revision.
>
> 2.  **The authors extend MEMIT editing algorithm, to the commonsense domain, by varying edit tokens and improving the layer selection strategy. The authors highlighted the results of doing so (line 88-103), but the reasons and intuition for doing so are not explained. It is better to explain why doing so before introducing the results.**
>
> **Intuition for varying edit tokens:** While factual knowledge depends mostly on the subject of a sentence, commonsense plausibility assessment relies on all components of a sentence. That's why we edited subject, verb and object tokens in our analysis. You can also find the intuition mentioned in Line 208-215 in the paper.\
> We **improved the layer selection strategy** in order to better align the causal tracing method and editing method. MEMIT chose to edit a five-layer window ending at the layer with the highest average indirect effect. However, all layers in the window span are equally important. Therefore, selecting the layers belonging to the window with the maximum moving average of the average indirect effect should only lead to better performance. Thus, we considered edit layers as a hyperparameter and searched among them. We will include this intuition in the camera-ready.
>
> We would be happy to clarify more questions or concerns.

---

### Official Review · Reviewer_bRLr · 2023-08-03

**Soundness:** 4

**Excitement:**

3: Ambivalent: It has merits (e.g., it reports state-of-the-art results, the idea is nice), but there are key weaknesses (e.g., it describes incremental work), and it can significantly benefit from another round of revision. However, I won't object to accepting it if my co-reviewers champion it.

**Paper Topic And Main Contributions:**

This paper focus on common sense knowledge editing in Transformer.
Whether common sense judgements are causally related to localised, editable parameters in Transformer is investigated and answered in the affirmative.
Through extensive experiments and analyses, the authors show a compelling future direction for incorporating feedback about common sense into Transformer through direct model editing.

**Questions For The Authors:**

1 Figures in the paper (e.g., Figure 1) should be briefly explained in the body of the paper, otherwise it is not clear exactly what the legend means.
2 How does each token in Figure 3 correspond to the word in the example sentence?
3 The moving average of AIE doesn't to be mentioned in the experimental section?  Suggest adding an analysis to show how it works.
4 What are the inputs and outputs when editing common sense  knowledge?
5 The text uses a binary classification task to implement editing, can the editing also be evaluated in the form of QA? Does the dichotomous form make editing less difficult.
6 Can a comparison with methods such as MEND be added to the experimental section?

**Reasons To Accept:**

The authors have carried out an in-depth study and analysis of editing on common-sense knowledge in Transformer, showing a strong causal relationship between Transformer's MLP layer and common-sense likelihood judgements.
These parameters can be directly edited for fixing common-sense errors.
The MEMIT parameter editing algorithm was refined for common-sense plausible predictions by changing the editing tokens and improving the layer selection strategy.
Through extensive experimentation and analysis, the authors show a compelling future direction for incorporating feedback about common sense into Transformer through direct model editing.

**Reasons To Reject:**

The method of the paper is very similar in nature to the MEMIT method and seems to be just applying the MEMIT method to the common sense dataset.

Compared to MEMIT just the number of modified tokens is higher and the causal tracking is unchanged accordingly.

**Reproducibility:**

4: Could mostly reproduce the results, but there may be some variation because of sample variance or minor variations in their interpretation of the protocol or method.

**Reviewer Confidence:**

4: Quite sure. I tried to check the important points carefully. It's unlikely, though conceivable, that I missed something that should affect my ratings.

---

> ### Author Rebuttal · Authors · 2023-08-29
>
> Thank you for appreciating that our paper is compelling, provides extensive experimentation and analysis, and focuses on an important problem! We address your one concern and questions below.
>
> **The method of the paper is very similar in nature to the MEMIT method and seems to be just applying the MEMIT method to the common sense dataset.
> Compared to MEMIT just the number of modified tokens is higher and the causal tracking is unchanged accordingly.**
>
> This is an **analysis paper** where we present comprehensive empirical results and discussion; our goal was **not** to propose a new model editing method. Prior works have exclusively focused on factual knowledge editing, we are the first to investigate whether the same methods can be applied to the more complicated and nuanced (but equally as important) domain of common sense knowledge. To this end, we conducted a **comprehensive study** of different aspects of MEMIT and its ability for such adaptation.
>
> To conduct a thorough analysis with the editing method, we created a **PROBE set specifically for commonsense**. The datasets include several new categories, i.e., "affected reasoning" and "affected neighborhood," on top of the traditional MEMIT style editing evaluation due to the more nuanced nature of commonsense. We also broadened the affected and unaffected neighborhood statements collection by replacing {S,V,O} and {S,O} respectively, instead of always replacing {S} as evaluated in MEMIT.
>
> **Editing factual knowledge vs Commonsense knowledge:** We elaborate on the unique challenges of commonsense knowledge editing as opposed to factual knowledge editing in lines 51-61. We provide further explanation below:
>
> 1.  While a model's factual knowledge can be assessed through next word prediction tasks, grasping commonsense knowledge is considerably more intricate, and relying solely on argmax next word predictions is insufficient to capture the extent of possible completions to a commonsense query. Hence, we transitioned our approach from the original next word prediction used in MEMIT to assessing sentence plausibility prediction.
>
> 2.  While transformer-based LMs typically excel at storing and predicting factual information, there remains a noticeable performance disparity when it comes to handling common sense.
>
> 3.  Moreover, commonsense knowledge is intricately interconnected, meaning that modifying one portion of this knowledge could considerably disrupt established accurate understandings. Consequently, evaluating the editing of common-sense knowledge necessitates a more extensive and thorough approach.
>
> To adapt to these challenges, we had to adapt the original MEMIT algorithm in several ways, including:
>
> 1.  Establishing that high performance is crucial to having a strong causal tracing result.
>
> 2.  Exploring different edit tokens (subject, verb, object) rather than a fixed choice(only subject in MEMIT).
>
> 3.  Treating edit layer as a hyperparameter, and improving selection of the range of layers to edit to be more aligned with causal tracing result.
>
> 4.  Tuned our edit strategies to balance efficacy (how many incorrect predictions are fixed) with minimum relapse (how much the edit hurts by changing  previously correct predictions). As previously mentioned, commonsense knowledge is intricately interconnected, and hence balancing efficacy and relapse is crucial.
>
> Only with these changes are we able to establish strong performance of direct model editing on the common sense plausibility domain.
>
> ### Questions For The Authors:
>
> **Q1: Figures in the paper (e.g., Figure 1) should be briefly explained in the body of the paper, otherwise it is not clear exactly what the legend means.**
>
> Thanks for the suggestion; we will add explanations for each figure under the corresponding sections.
>
> **Q2: How does each token in Figure 3 correspond to the word in the example sentence?**
>
> Figure 3 displays the **average indirect effect** for all statements in PEP3k Edit Validation Set when subject, verb and object tokens are corrupted individually. For a statement such as "Soil absorbs oil" the subject is "soil", the verb is "absorbs" and the object is "oil". This sentence is tokenized as ['So', 'il', ' absorbs', ' oil']. Based on the {s,v,o} labels in a statement, we identify the first and last token after tokenization and measure their indirect effects. In this case, the first subject token is 'So', last subject token is 'il', both first and last verb token is ' absorbs', and finally both first and last object token is ' oil'.  After calculating the IE for each statement and its tokens, we plot the average values (AIE) as the causal tracing heat maps in Figure 3.
>
> **Q3: The moving average of AIE doesn't seem to be mentioned in the experimental section? Suggest adding an analysis to show how it works.**
>
> Thank you for pointing this out.
>
> Here is a description, which we will add given extra space in the next version:
>
> For demonstration purposes let's assume there exists a model with just 10 layers. The average indirect effects of these layers at our desired edit token (let's assume last verb token) are: [0.0, 0.1, 0.2, 0.3, 0.5, 0.4, 0.4, 0.3, 0.2, 0,0]
>
> Let's also assume that we are considering only 5 layer windows. The highest average indirect effect is observed at the 5th layer with value 0.5. According to MEMIT, the optimal edit layers will be a 5 layer window ending at the highest AIE layer, in this case it will be the layers 1,2,3,4,5.
>
> Now let's calculate the moving average of 5 layer windows. The moving average of layers 1-5 is (0.0 + 0.1 + 0.2 + 0.3 + 0.5) / 5 = 0.22, similarly the moving average of layers 2-6 will be (0.1 + 0.2 + 0.3 + 0.5 + 0.4) / 5 = 0.3 and so on. The moving averages of all 5 layer windows are: [0.22, 0.3, 0.36, 0.38, 0.36, 0.26]
>
> The maximum moving average is observed for layers 4-8 with value 0.38. In our method, we would also consider layers 4-8 along with MEMIT's layers 1-5.
>
> During hyper-parameter tuning, using results similar to the below table, we compare windows within the search space and pick the best performing strategy for {s,v,o} tokens. In comparison, MEMIT does not explore the layer space and picks the 5 layers prior to the layer with max AIE for the subject token (**Table 7 in our paper**).
>
> **Table:** Layers with max average IE and max moving average IE for the PEP3k Edit Validation Set
>
> | Model | Edit Token | Layer with Max AIE | Layers (size=3) with Max Moving Avg IE | Layers (size=5) with Max Moving Avg IE |
> |---|---|---|---|---|
> | GPT-2 Large | Last Subject | 8 | 8,9,10 | 8,9,10,11,12 |
> | GPT-2 Large | Last Verb | 4 | 4,5,6 | 4,5,6,7,8 |
> | GPT-2 Large | Last Object | 12 | 11,12,13 | 10,11,12,13,14 |
> | GPT-2 XL | Last Subject | 5 | 4,5,6 | 2,3,4,5,6 |
> | GPT-2 XL | Last Verb | 5 | 5,6,7 | 3,4,5,6,7 |
> | GPT-2 XL | Last Object | 12 | 10,11,12 | 9,10,11,12,13 |
>
>
> In this table, we observe that the moving average method results in a different set of layers than MEMIT's method in all cases. We will include this analysis in the experiments section of the main paper in the next version.
>
> **Q4: What are the inputs and outputs when editing common sense knowledge?**
>
> If you mean the formats of the natural language query and expected output, we describe this in lines 184-186:
>
> The task is to predict a label y ∈ {True, False} given an input triple x = (s, v, o).
>
> If you mean the overall algorithm:
>
> Inputs to edit a statement:
>
> -   Transformer Language Model
>
> -   Statement of format (s,v,o) with incorrect plausibility P (True/False)
>
> -   The target plausibility i.e.  P' (False/True)
>
> -   Editing Token ∈ {s,v,o}
>
> -   Layers to Edit
>
> Output:
>
> -   Edited Transformer Language Model
>
> Let us know if this is still confusing.
>
> **Q5: The text uses a binary classification task to implement editing, can the editing also be evaluated in the form of QA? Does the dichotomous form make editing less difficult?**
>
> We chose to model the problem statement as the binary plausibility of the sentence instead of using a QA format as commonsense questions can have multiple correct answers. This choice can be used to evaluate a variety of statements such as, "Humans eat" : "vegetable", "milk", "meat" and many other foods all of which are considered plausibly 'True' in the (s,v,o) format. We hope this answers your question, we'll be happy to clarify this further.
>
> **Q6: Can a comparison with methods such as MEND be added to the experimental section?**
>
> According to the results in MEMIT ([Meng et al., 2023](https://arxiv.org/pdf/2210.07229.pdf)) Table 2, other editing methods such as ROME and MEND perform poorly compared to MEMIT by 35.5% and 62.7% respectively. Therefore, we mainly focused on MEMIT and compared our approach with MEMIT in Table 2 of our paper.
>
> We would be happy to clarify more questions or concerns.

---

### Official Review · Reviewer_wc3N · 2023-08-05

**Soundness:** 4

**Excitement:**

2: Mediocre: This paper makes marginal contributions (vs non-contemporaneous work), so I would rather not see it in the conference.

**Paper Topic And Main Contributions:**

This paper mainly discusses whether the method memit, previously used for editing factual knowledge, can be used to modify erroneous commonsense in models. The paper explores the locating of commonsense in language models and the editing issues associated with it. It discovers that commonsense is stored in specific layers of language models and can be modified using previous model editing methods. Experimental results indicate the effectiveness of the proposed method, memit-csk, in editing commonsense.

**Reasons To Accept:**

1. This paper presents a rich set of experiments that explore knowledge editing issues with respect to commonsense. The discussions on the problem itself are also comprehensive, delving into configuration generalization and semantic generalization in the methods.

2. The writing and presentation of the paper is clear.

3. The authors promise to provide the code for readers to replicate their findings.

**Reasons To Reject:**

1. The paper should discuss more about the connection and distinction between editing factual knowledge and commonsense knowledge. The current version appears to be very similar to Rome and memit, and thus seems like an incremental work.

2. The previous memit method has been proven to be applicable for editing models of around 20 billion parameters. This paper only considers GPT-2 Large and XL, and it would be beneficial to explore commonsense editing for larger models.

3. The abstract mentions that memit can be used to update black-box models, but both knowledge locating and editing require probability distribution and gradient information from the model, which contradicts the claim of it being a black-box approach.

**Reproducibility:**

4: Could mostly reproduce the results, but there may be some variation because of sample variance or minor variations in their interpretation of the protocol or method.

**Reviewer Confidence:**

5: Positive that my evaluation is correct. I read the paper very carefully and I am very familiar with related work.

---

> ### Author Rebuttal · Authors · 2023-08-29
>
> Thank you for appreciating our rich set of experiments, comprehensive discussion, clear writing & presentation, and the strength of our claims/arguments! We address your concerns below.
>
> 1\. **The paper should discuss more about the connection and distinction between editing factual knowledge and commonsense knowledge. The current version appears to be very similar to Rome and memit, and thus seems like an incremental work.**
>
> **Distinctions between editing factual knowledge and commonsense knowledge:** We elaborate on the unique challenges of commonsense knowledge editing as opposed to factual knowledge editing in lines 51-61, but agree with you that more discussion will be worthwhile in the camera-ready. We provide further explanation below:
>
> 1.  While a model's factual knowledge can be assessed through next word prediction tasks, grasping commonsense knowledge is considerably more intricate, and relying solely on argmax next word predictions is insufficient to capture the extent of possible completions to a commonsense query. Hence, we transitioned our approach from the original next word prediction used in MEMIT to assessing sentence plausibility prediction.
>
> 2.  While transformer-based LMs typically excel at storing and predicting factual information, there remains a noticeable performance disparity when it comes to handling common sense.
>
> 3.  Moreover, commonsense knowledge is intricately interconnected, meaning that modifying one portion of this knowledge could considerably disrupt established accurate understandings. Consequently, evaluating the editing of common-sense knowledge necessitates a more extensive and thorough approach.
>
> To adapt to these challenges, we had to adapt the original MEMIT algorithm in several ways, including:
>
> 1.  Establishing that high performance is crucial to having a strong causal tracing result.
>
> 2.  Exploring different edit tokens (subject, verb, object) rather than a fixed choice(only subject in MEMIT).
>
> 3.  Treating edit layer as a hyperparameter, and improving selection of the range of layers to edit to be more aligned with causal tracing result.
>
> 4.  Tuned our edit strategies to balance efficacy (how many incorrect predictions are fixed) with minimum relapse (how much the edit hurts by changing  previously correct predictions). As previously mentioned, commonsense knowledge is intricately interconnected, and hence balancing efficacy and relapse is crucial.
>
> Only with these changes are we able to establish strong performance of direct model editing on the common sense plausibility domain.
>
> **The current version appears to be very similar to Rome and memit, and thus seems like an incremental work:** This is an **analysis** paper where we present comprehensive empirical results and discussion; our goal was **not** to propose a new model editing method. Prior works have exclusively focused on factual knowledge editing, we are the first to investigate whether the same methods can be applied to the more complicated and nuanced (but equally as important) domain of common sense knowledge. To this end, we conducted a **comprehensive study** of different aspects of MEMIT and its ability for such adaptation.\
> To conduct a thorough analysis with the editing method, we created a **PROBE set specifically for commonsense**. The datasets include several new categories, i.e., "affected reasoning" and "affected neighborhood," on top of the traditional MEMIT style editing evaluation due to the more nuanced nature of commonsense. We also broadened the affected and unaffected neighborhood statements collection by replacing {S,V,O} and {S,O} respectively, instead of always replacing {S} as evaluated in MEMIT.
>
> 2\. **The previous memit method has been proven to be applicable for editing models of around 20 billion parameters. This paper only considers GPT-2 Large and XL, and it would be beneficial to explore commonsense editing for larger models.**
>
> Although we were interested in experimenting with larger models, GPU resource constraints limited us to GPT-2 XL. MEMIT requires lots of memory to run--as mentioned in the reproducibility section of the original paper (Meng et al. 2022), "Due to the size of these language models, our experiments will not run on GPUs with less memory [than 48GB for GPT-J and ~96GB for GPT-Neo-X]". Additionally, commonsense editing first requires fully fine-tuning the model on the task (sections 3.1, 5.1), which consumes significantly more memory. We suspect the editing success trend with larger models will likely be similar to what we observed with GPT2-L and GPT2-XL given the scaling success of MEMIT reported by Meng et al., and we leave this as future work for verification.
>
> 3\. **The abstract mentions that memit can be used to update black-box models, but both knowledge locating and editing require probability distribution and gradient information from the model, which contradicts the claim of it being a black-box approach.**
>
> Thanks for catching this and we are sorry for the confusion. It should have been "open-source transformer-based" models instead of "black-box" models. We will correct this in the updated version.
>
> We would be happy to clarify more questions or concerns.

---

### Meta-Review · Area_Chair_sRM9 · 2023-09-19

**Recommendation:** 5

**Metareview:**

This paper presents a study on editing the issues in editing common-sense knowledge in transformers.

The study is well-motivated. While there has been work in understanding and applying edits to factual knowledge, the issues involved in editing common-sense knowledge are largely unknown. The paper presents a high-quality and well-designed study by suitably adapting an existing method. The paper is well written. The main significance of the work is in the findings of the study, rather than in the modeling aspect. The other significant result from the paper are the datasets that have been created as part of the study.
The problem of editing complex inter-linked knowledge such as common-sense is an important one and understanding aspects of this problem is crucial.

In terms of weakness, the writing can be improved to more clearly present the key adaptations that have been made, so that the contributions relative to existing methods are clearer. This however is a minor issue that can be fixed easily.

---

### Decision · Program_Chairs · 2023-10-07

**Decision:**

Accept-Main

**Comment:**

This paper presents a study on editing the issues in editing common-sense knowledge in transformers.

The study is well-motivated. While there has been work in understanding and applying edits to factual knowledge, the issues involved in editing common-sense knowledge are largely unknown. The paper presents a high-quality and well-designed study by suitably adapting an existing method. The paper is well written. The main significance of the work is in the findings of the study, rather than in the modeling aspect. The other significant result from the paper are the datasets that have been created as part of the study.
The problem of editing complex inter-linked knowledge such as common-sense is an important one and understanding aspects of this problem is crucial.

In terms of weakness, the writing can be improved to more clearly present the key adaptations that have been made, so that the contributions relative to existing methods are clearer. This however is a minor issue that can be fixed easily.